# Assessing the relationship between menstrual products and reproductive and urogenital tract infections (RUTIs): A systematic review evaluating the evidence and recommendations for future research

Ana Krause[1]*, Gregory K. Zane[2], Andrea C. Rivas-Nieto[2], Abigail Mulugeta[1], Md. Rezaul Hossain[2], R. Scott McClelland[1,2,3], Jairam R. Lingappa[1,4], Stephen E. Hawes[1,2]

1 Department of Global Health, School of Public Health, University of Washington, Seattle, Washington, United States of America, 2 Department of Epidemiology, School of Public Health, University of Washington, Seattle, Washington, United States of America, 3 Department of Medicine, Division of Allergy and Infectious Diseases, University of Washington, Seattle, Washington, United States of America, 4 Department of Pediatrics, School of Medicine, University of Washington, Seattle, Washington, United States of America

* akrause2@uw.edu

## Abstract

### Background

Concerns regarding the effects of non-tampon menstrual products on reproductive and urogenital health, particularly the risk of infections, is an area of ongoing investigation. We conducted an updated systematic review to assess the methodological quality of current evidence assessing associations between menstrual product use and reproductive and urogenital tract infections (RUTIs), and offer recommendations for future research.

### Methods

Three databases (PubMed, Web of Science, United States Food and Drug Administration Manufacturer User Facility Device Experience) were searched for relevant published studies or product safety reports up to October 13, 2024. We included studies on menstruators of any age and geography assessing for any reusable/disposable menstrual pads, menstrual cups, or homemade alternatives worn only for menstrual absorbency compared to other menstrual products or no product use, with outcomes centered on RUTIs. Protocols, reviews, and studies assessing only tampons or non-menstrual absorbents were excluded. Results were evaluated and synthesized using tabular methods according to measures of association, and assessed across four criteria categories 1) product definition, 2) comparator definition, 3) outcome definition, and 4) confounder consideration.

**Data availability statement:** All relevant data are within the paper and its Supporting Information files.

**Funding:** This work was supported by the Gates Foundation as part of the University of Washington START Center grant No. GR018228. The funder commissioned the study but did not have control over the design, data collection, analysis, conclusions, or preparation of the manuscript.

**Competing interests:** RSM received research support and STI testing supplies, unrelated to the current research, from Hologic Corporation. SEH is funded by grants from the U.S. National Institutes of Health and the Gates Foundation. GKZ has received a grant from the U.S. National Institutes of Health, unrelated to this work, and has received consulting employment through Boston Scientific Corporation, unrelated to this work. JL had received grants from the U.S. National Institutes of Health, unrelated to this work, and is on the Scientific Advisory Board for and received stock in Prosetta Bioconformatics, Inc. AK has received consulting employment through the Gates Foundation, unrelated to this work. There are no patents, products in development or marketed products associated with this research to declare. This does not alter our adherence to PLOS ONE policies on sharing data and materials.

## Results

Thirty-one studies were included in this review. Most studies clearly defined outcomes and considered necessary confounders. In contrast, studies with well-defined products (6.5%) and comparator products (9.7%) were uncommon. Just 3.2% and 9.7% of studies fully defined products and comparators, respectively. Ten studies (32.2%) reported some data on four or more confounders, and seventeen (54.8%) defined their infectious outcomes and included laboratory confirmation. A meta-analysis was not possible due to data heterogeneity across product, comparator, and outcome definitions. Overall, associations between menstrual products and RUTIs are inconclusive.

## Conclusion

Future studies should 1) clearly define product and comparator type, material, frequency of change, and washing, drying, and storage practices for reusable products, 2) prioritize laboratory or clinician-confirmed outcomes over self-reported symptoms, and 3) adjust for relevant confounders.

---

## Introduction

Over the course of their lifetime, menstruators will spend nearly four decades managing menstruation, experiencing over 450 cycles and using upwards of 10,000 menstrual products per individual [1]. Menstrual products encompass a range of absorbent materials including those made explicitly for the purpose of menstrual absorbency, such as sanitary pads, tampons, menstrual cups, and period underwear but can also include makeshift materials such as repurposed cloths/rags, toilet paper, newspaper, cotton wool, leaves, chicken feathers, or mud [1–3]. While access to and affordability of menstrual products are core tenets of menstrual health and hygiene management, the safety of non-tampon products, especially for populations with inadequate water, sanitation, and hygiene (WaSH) access, remains an area of ongoing investigation and discussion. Concerns about the impact of different menstrual products and materials on menstruators' reproductive and urogenital health, including risks of infections, and health, social, and educational outcomes, are frequently discussed within the context of period poverty [1,4,5]. However, rigorous understanding of the potential health implications of using different menstrual products have been limited in part by a lack of high-quality intervention studies and heterogenous data [4,6,7]. A 2013 systematic review and meta-analysis on menstrual hygiene management identified a need for "methodologically consistent research" citing wide variations in exposure definitions, nonspecific outcome measures, and insufficient adjustment for relevant confounders [4]. However, that review and others did not systematically assess the inclusion of relevant behavioural risk factors, menstrual product factors, and outcome measurements in their analyses, further limiting understanding of specific menstrual product associated risk.

We therefore sought to conduct an updated systematic search and review to assess the methodological quality of current evidence assessing associations between menstrual product use and reproductive and urogenital tract infections (RUTIs), identify methodological gaps in the literature, and provide key recommendations for future research.

## Methods

### Eligibility criteria

Our population, intervention/exposure, comparator, and outcome (PICO) criteria were: P – menstruators of any age group, located in any geography; I – reusable or disposable menstrual pads, menstrual cups, and homemade alternatives (e.g., cloths/rags) all worn only for the purpose of menstrual absorbency; C – other menstrual products or none; O – RUTIs including human papillomavirus (HPV), human immunodeficiency virus (HIV), syphilis, gonorrhea, chlamydia, bacterial vaginosis (BV), urinary tract infections (UTI), vaginitis, and candidiasis. We excluded protocols, reviews, non-peer-reviewed studies, and studies that assessed only tampons or non-menstrual absorbents (e.g., liners for urinary incontinence). Eligible studies included any peer-reviewed study type except protocols or reviews. Studies in all languages were considered based on the team's language skills and assistance from artificial intelligence (AI) (ChatGPT-4) for specific articles [8].

### Search strategy

We searched three databases (PubMed, Web of Science, and the United States Food and Drug Administration Manufacturer and User Facility Device Experience – FDA MAUDE) for safety reports and relevant studies published globally up to October 13, 2024 (see S1 Table for the full search strategy). FDA MAUDE was included to determine whether any product related safety reports had occurred since a 2013 systematic review [4]. Non-English articles were reviewed by team members with relevant language expertise. For the single article with an available English abstract but with our team lacking a member fluent in the language of the full text, AI tools (ChatGPT-4) were employed to assist with translation for full-text review [9].

### Screening and data extraction

Covidence [10] was used to manage search results and screen articles for eligibility based on inclusion and exclusion criteria. Two reviewers screened each article and independently assessed each full text for eligibility. Conflicts were resolved through reviewer discussion. Information on article metadata, study methods (aim(s), design, data collection period, inclusion/exclusion criteria, sample size, population characteristics including age and relevant confounding variables), and exposures and outcomes (included menstrual products, comparators, any eligible outcome measures, results) were extracted using Covidence (see S2 Appendix for Data Extraction Spreadsheet). Each article was extracted by an independent reviewer and checked by a second team member.

### Quality assessment and analysis

Studies were assessed for quality by two reviewers independently across four criteria: 1) product definition, 2) comparator definition, 3) outcome definition, and 4) confounder consideration. Each criterion was scored from two to five stars based on the completeness of its definitions across relevant factors. For the product definition, four factors were considered: i) product type beyond a label of "pad", "cloth", or "menstrual cup", including details such as commercially made disposable, homemade reusable, heavy absorbent, panty liner, or product brand name, ii) product material (e.g., silicone, cotton, nylon, silk, perfumed/non-perfumed, etc.), iii) frequency of product change, and iv) any information on washing/drying/storage practices for reusable products. For the comparator definition, two factors were considered, including whether the comparator product was defined or not (according to at least product type, material, and frequency of product change) or

if "usual or traditional practices" were defined. For the outcome definition, two factors were considered, including whether the study authors defined their RUTI outcome(s) and whether laboratory testing (self-testing or by a trained health worker) was conducted. For confounders, five categories were considered, including whether any data was reported about study participants: a) sexual behaviour(s), ii) socioeconomic status, iii) WaSH including personal hygiene practices (e.g., douching, etc.), iv) age, and v) recent antibiotic use. Effects of included studies were summarized and presented using tabular methods only detailing study characteristics and results. A meta-analysis to assess risk of RUTIs outcomes by product was planned if feasible; otherwise we synthesized findings according to the Synthesis Without Meta-analysis (SWiM) guideline for reviews where statistical combination of results is not possible [11], focusing on the product, comparator, outcome, and confounder definition gaps identified during our quality assessment (see S3 Appendix for SWiM and PRISMA Checklists). Studies were scored, one to four stars, based on partial or full definitions across the four quality criteria with each category weighted equally (see Table 2 footnote, S4 Table for additional details). We did not include weighting of different criteria in our quality assessment because the purpose was to conduct a descriptive evaluation of study quality rather than to generate a weighted score. This approach aligns with common practices where quality assessments serve to characterize methodological strengths and limitations without imposing arbitrary weights. A full breakdown of the quality assessment results, excluding any cut-offs or summary scores is available in S4 Table for transparency. This approach aligns with a previous systematic review that synthesized findings using a specific criterion quality assessment [12]. This review was not registered.

## Results

### Search results

Our search identified 811 articles (461 from PubMed and 350 from Web of Science) with 438 duplicates removed by Covidence or manually by the study team. No safety reports through the U.S. FDA MAUDE database were identified. Following title and abstract screening of 373 articles, 272 were deemed ineligible as they did not meet the PICO criteria. One hundred and one studies proceeded to full-text review with 70 excluded for having the wrong outcomes (not RUTIs) (33), wrong product exposure (other than non-tampon menstrual products) (27), wrong study design (e.g., protocol or a review not evaluating the exposure/outcome of interest) (8), wrong patient population (i.e., non-menstruators), or an abstract/poster with no corresponding publication (1) (Fig 1). Thirty-one studies were deemed eligible and included in this review (Table 1). A meta-analysis of outcomes was not possible due to data heterogeneity across the product, comparator, and outcome definitions as shown in Table 2. Included studies were conducted in lower-middle- (n = 15), high- (n = 11), lower- (n = 4), and upper-middle-income countries (n = 1), with India (n = 8), Kenya (n = 5), and the United States (n = 5) most often cited.

### Product definition: Criterion 1

Of the four product factors considered in our analysis (product type, material, frequency of product change, and washing/drying/storage practices for reusable products), only one article provided partial (3.2%) and one other article (3.2%) provided full definitions across all four factors (Table 2) [13,14]. An additional 10 and six articles partially or fully defined menstrual products across three and two factors respectively. Thirteen articles (41.9%) did not define the menstrual product(s) beyond basic labels of "pad" or "cloth" limiting more nuanced understanding of potential product-associated risks. Only eight articles provided any product material information and just three did so for both intervention and control groups [13,15,16]. Some studies reported use of "other makeshift items" or "usual practice" without offering further details [17–20]. Despite infrequent changing of menstrual products being understood as a risk factor for some RUTI outcomes [7,21,22], only 11 articles (35.5%) captured any data on this aspect, with two of these studies not considering frequency data in their analysis [23,24]. Although 10 articles specified the inclusion of reusable products, only three captured data

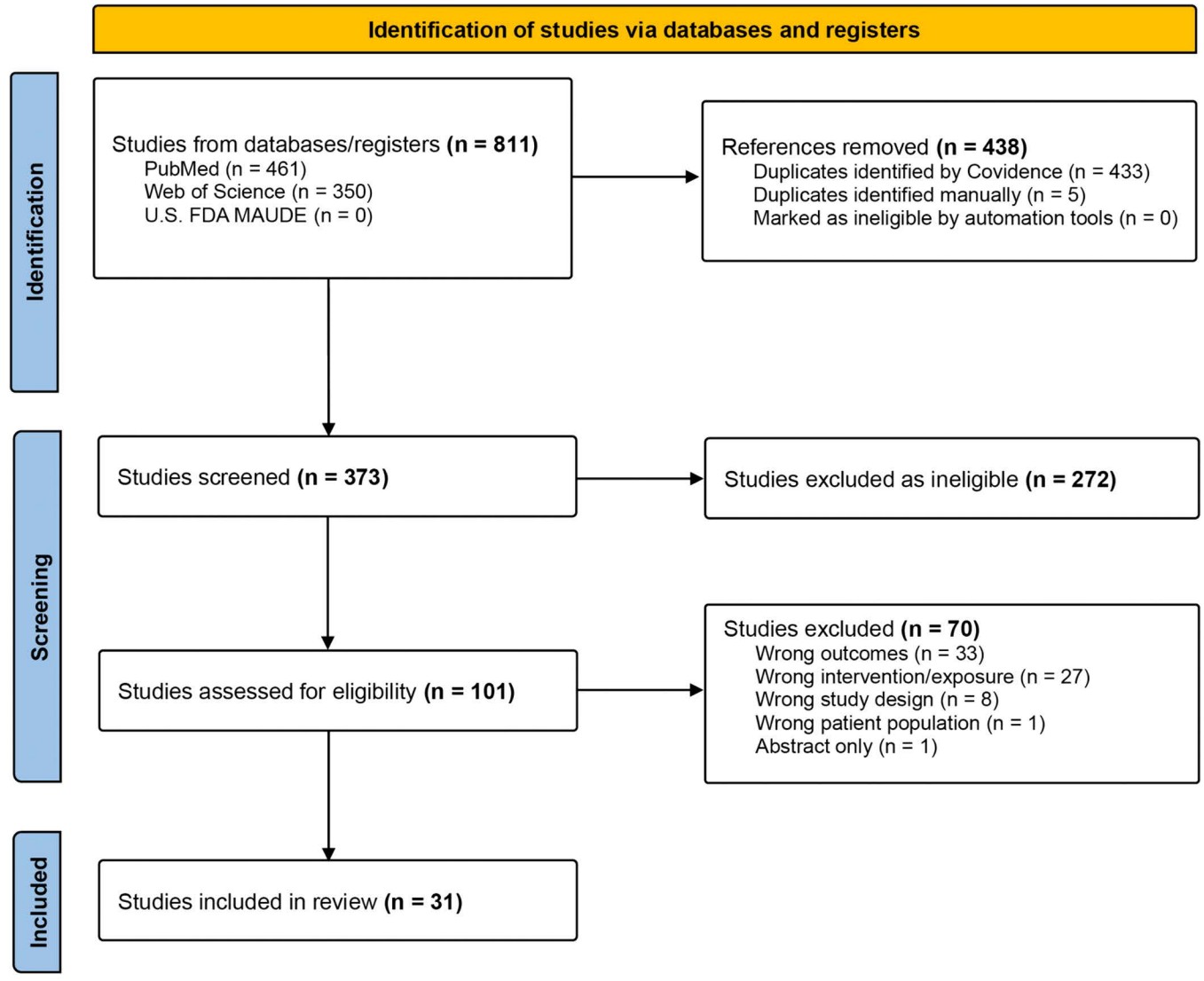

**Fig 1. PRISMA Flowchart.**

on washing, drying, and storage practices [13,14,16], while another three studies reported on just one practice [24–26]. Due to products being poorly defined, it was unclear in most studies whether the "pad" and/or "cloth" was reusable or disposable. Most studies assigned participants to a single product category, with seven labeling these as a single product only group despite the combined use of products being reported in the text or supplementary data. An example of this is Zulaika's 2023 cluster randomized controlled trial (cRCT), where 82–93% of girls in the "cup group" were also using pads, along with cloths, tampons and other undefined products at study baseline, mid-point, and end [20].

## Comparator definition: Criterion 2

Just three studies (9.7%) fully defined their comparator product type across the four product factors [13,14,20]. Another five partially defined their comparator and 17 provided a product category (e.g., "pad") but no additional information. Six studies, including four published since 2022, did not define their comparator relying on terms such as "usual/traditional

**Table 1. Study Characteristics, listed chronologically (N = 31 studies).**

| Study | Design | N[1] | Geography[2] | Study Population |
|---|---|---|---|---|
| Leibovici, 1984 | Cross-sectional | 147 | Israel[H] | Women aged 17–34 years with bacteriological evidence of UTIs |
| Hansen, 1985 | Cross-sectional | 509 | Denmark[H] | Care-seeking, women of reproductive age |
| Foxman, 1995 | Case-control | 374 | United States[H] | Adult women using university health services without a history of UTIs |
| Geiger, 1996 | Case-control | 495 | United States[H] | Non-pregnant female university students |
| Omar, 1998 | Cross-sectional | 250 | United States[H] | Sexually active female college students aged 17–21 years |
| Tchoudomirova, 1998 | Case-control | 927 | Sweden[H] | Contraceptive attendees at family planning/youth clinics (15–44 years) |
| Demba, 2005 | Cross-sectional | 227 | The Gambia[L] | Non-pregnant women with self-reported vaginal symptoms (18–50 years) |
| Morison, 2005 | Quasi-experimental | 30 | The Gambia[L] | Married, regularly menstruating females aged 20–53 years |
| Unzeitig, 2007 | Prospective study | 100 | Czechia[H] | Sexually active females with regular menstrual cycles |
| Baisley, 2009 | Cross-sectional | 1305 | Tanzania[LM] | Non-pregnant, HSV-2 sero-positive women aged 16–35 years |
| Klatt, 2010 | Case-control | 140 | United States[H] | Adult women |
| Klebanoff, 2010 | Cross-sectional | 3620 | United States[H] | Non-pregnant women aged 15–44 years presenting for routine health care visits |
| Howard, 2011 | Randomized controlled trial | 89 | Canada[H] | Non-pregnant women with previous tampon use (19–40 years) |
| Balamurugan, 2012 | Cross-sectional | 656 | India[LM] | Women aged 15–45 years |
| Philip, 2013 | Cross-sectional | 260 | India[LM] | Married women aged 15–44 years |
| Das, 2015 | Case-control | 486 | India[LM] | Care-seeking, non-pregnant women of menstruating age (18–45 years) |
| Phillips-Howard, 2015 | Cross-sectional | 2112 | Kenya[LM] | Menstruating females 13–29 years of age living in rural areas |
| Phillips-Howard, 2016 | Randomized controlled pilot | 644 | Kenya[LM] | Primary schoolgirls with 3 or more experienced menses (14–16 years) |
| Abulizi, 2017 | Cross-sectional | 6000 | China[UM] | Non-pregnant, sexually active rural women aged 21–60 years |
| Torondel, 2018 | Cross-sectional | 558 | India[LM] | Non-pregnant women aged 18–45 years attending two public hospitals |
| Janoowalla, 2020 | Prospective cohort | 209 | Rwanda[L] | Menstruating female secondary school students aged 18–24 years |
| Das, 2021 | Cross-sectional | 1164 | India[LM] | Non-pregnant women of reproductive age living without HIV (18–45 years) |
| Mehta, 2021 | Cross-sectional | 436 | Kenya[LM] | Adolescent schoolgirls aged 14–22 years |
| Nabwera, 2021 | Cross-sectional | 358 | The Gambia[L] | Non-pregnant, menstruation-experienced schoolgirls (15–21 years) |
| Singh, A 2022 | Cross-sectional | 629 | India[LM] | Adolescent schoolgirls aged 14–19 years |
| Singh, M 2022 | Cross-sectional | 149 | India[LM] | Ever-married, insured women living in urban slum (15–49 years) |
| Chakrabarty, 2023 | Cross-sectional | 27983 | India[LM] | National sample of women aged 15–24 years with prior knowledge of STIs |
| Mehta, 2023 | Nested longitudinal analysis | 436 | Kenya[LM] | Adolescent schoolgirls aged 14–22 years |
| Zulaika, 2023 | Randomized controlled trial | 4106 | Kenya[LM] | Non-pregnant adolescent schoolgirls with 3 or more experienced menses |

*(Continued)*

**Table 1.** (Continued)

| Study | Design | N[1] | Geography[2] | Study Population |
|---|---|---|---|---|
| Al Karmi, 2024 | Cross-sectional | 387 | Jordan(LM) | Non-pregnant, adult women residing in refugee camps aged 18–49 years |
| Madar, 2024 | Prospective cohort | 103 | France(H) | Adult women consulting for IUD insertion or change |

[1]To simplify study details, the reported study population size (N) reflects the total number of participants used to assess the association between menstrual health products and infectious disease outcomes. It does not reflect stratified subpopulations based on specific products or disease outcomes.

[2]Includes country-specific World Bank income-level classification for 2024–2025: (H) = High income; (UM) = Upper-middle income; (LM) = Lower-middle income; (L) = Low income.

Abbreviations: UTI, urinary tract infection; HSV-2, herpes simplex virus-2; STI, sexually transmitted infections; HIV, human immunodeficiency virus; IUD, intrauterine device

practice" or "other types" [18,19,23,24,27,28]. Poorly defined comparison groups greatly diminish our ability to accurately assess the relationship(s) between menstrual products and RUTI outcomes.

## Outcome definition: Criterion 3

Over half (17/31) of studies defined their RUTI outcomes either by the specific bacterial, viral, parasitic, or fungal outcome or category (e.g., UTI) and included laboratory confirmation either by a trained health worker or self-testing. Five studies included a partial outcome definition or laboratory testing for some study groups (e.g., clinic controls but not population controls, etc. [29]) or some outcomes (e.g., symptoms for RTI, but urine dipstick for UTI [26]). Four studies either defined their outcomes or used some form of laboratory testing, and five studies did not define their outcomes and relied on self-reported symptoms [15,27,30–32].

## Confounder consideration: Criterion 4

Of the five confounder categories that were considered (sexual behaviours, socioeconomic status, WaSH including personal hygiene practices and vaginal douching, age, and recent antibiotic use), less than a third (10/31) of studies reported some data on four or more confounders and/or factored them in as part of their study inclusion/exclusion criteria (e.g., women using antibiotics in the previous three weeks were excluded). Age was the most common confounder considered by 27 of 31 studies, however two of these studies did not adjust for age in their statistical models despite reported statistically significant differences in outcome prevalence between different age groups [23,31]. Two thirds of studies (21/31) reported some sexual behaviour data or excluded sexually inactive women. However, the amount of sexual behaviour data varied by study, with ten studies reporting limited data such as solely categorizing menstruators as sexually active or not [20,25,32,33] and/or capturing data on number of partners [15,24,28,34], frequency of sex [35], or pregnancy history [17]. Other studies with more detailed sexual behaviour data reported on frequency of masturbation/penetrative/oral sex, and/or use of contraceptive methods including barriers or spermicide, menstrual sex, sexual violence, and/or transactional sex for menstrual products [17,19,29,36–41]. Multiple studies assessing RTIs/STIs did not capture any sexual practice data with several authors indicating that it would have been considered culturally inappropriate to capture this information [14,23,26,27,31,42]. Lastly, two studies did not include sexual behaviours in their conceptual model, however Torondel highlighted the role of sex in BV within their article [13,14]. Less than half of studies (14/31) considered any personal hygiene practices such as vaginal douching, with one study only capturing hygiene data related to handwashing practices [24]. Recent antibiotic use was only considered by eight studies, half of which excluded women with a history of antibiotic use in the previous two to three weeks [13,15,16,43].

In sum, of the four criteria, studies demonstrated greater rigor in defining outcomes and considering confounders. In contrast, products and comparators remained largely ill-defined (Table 2).

**Table 2. Appraisal of Quality of Evidence, studies listed chronologically (N = 31).**

| Study (First Author, Year) | Outcome | Evidence for each criterion | | | |
|---|---|---|---|---|---|
| | | Criterion 1: Product Definition | Criterion 2: Comparator Definition | Criterion 3: Outcome Definition | Criterion 4: Confounder Consideration |
| Leibovici, 1984 | UTI | * | ** | **** | ** |
| Hansen, 1985 | GV | * | * | **** | * |
| Foxman, 1995 | UTI (first time) | * | ** | **** | ** |
| Geiger, 1996 | Candidiasis | * | *** | *** | **** |
| Omar, 1998 | UTI, Gonorrhea, Chlamydia, Syphilis, HIV | * | ** | ** | ** |
| Tchoudomirova, 1998 | UTI (recurrent) | * | ** | **** | ** |
| Demba, 2005 | BV, HIV | ** | *** | **** | **** |
| Morison, 2005 | BV | *** | ** | **** | ** |
| Unzeitig, 2007 | BV, UTI | * | ** | **** | * |
| Baisley, 2009 | BV | ** | *** | **** | *** |
| Klatt, 2010 | BV (recurrent) | * | ** | *** | ** |
| Klebanoff, 2010 | BV | * | ** | **** | **** |
| Howard, 2011 | UTI, Vaginitis | *** | ** | * | *** |
| Balamurugan, 2012 | RTI unspecified | * | ** | * | ** |
| Philip, 2013 | RTI/STI unspecified | * | ** | * | ** |
| Das, 2015 | BV & UTI | *** | *** | *** | **** |
| Phillips-Howard, 2015 | HIV | *** | ** | **** | **** |
| Phillips-Howard, 2016 | BV, Candida Albicans, Trichomonas, Gonorrhea, Chlamydia | *** | * | **** | *** |
| Abulizi, 2017 | HPV | * | ** | **** | **** |
| Torondel, 2018 | Candida, BV, Trichomonas | **** | **** | **** | **** |
| Janoowalla, 2020 | UTI & VV symptoms | *** | *** | *** | **** |
| Das, 2021 | BV & VVC | **** | **** | **** | *** |
| Mehta, 2021 | BV, STI | * | ** | **** | **** |
| Nabwera, 2021 | UTI, BV & Candida | *** | ** | *** | *** |
| Singh, A 2022 | RTI/STI unspecified | ** | * | * | *** |
| Singh, M 2022 | Gonorrhea, Chlamydia chancroid, & HIV | ** | ** | ** | ** |
| Chakrabarty, 2023 | RTI unspecified | ** | ** | * | *** |
| Mehta, 2023 | BV, Trichomonas, Gonorrhea, Chlamydia | *** | * | **** | **** |
| Zulaika, 2023 | HIV | *** | **** | **** | *** |
| Al Karmi, 2024 | UTI; RTI unspecified | ** | * | ** | *** |
| Madar, 2024 | Urogenital infection | *** | * | ** | *** |

**Product Definition:** ****: Product(s) partially or fully defined for ≥4 factors: product type, product material, frequency of product change, washing/drying/storage practices for reusable products. ***: Product(s) partially or fully defined for 3 factors, **: product partially or fully defined for 1–2 factors, *: product(s) not defined beyond label.

**Comparator Definition:** ****: defined (according to at least the first 3 product factors), ***: not fully defined, but more information than product category alone, **: compared defined by product category (e.g., pad) but no further product information, *: not defined (e.g., "usual practice" with no further information)

**Outcome Definition:** ****: outcome(s) defined for all groups & confirmed with laboratory testing (self-testing or trained health worker), ***: either some/all outcome(s) not fully defined, or laboratory testing only used on some study groups but not all, **: either outcome defined, or confirmatory laboratory testing used (self-testing or trained health worker), *: outcome(s) not defined & no confirmatory laboratory testing used (i.e., symptom self-report)

**Confounder Definition:** ***: ≥4 relevant confounders included re; i) sexual behaviour(s), ii) socioeconomic status, iii) WASH including personal hygiene practices, iv) age, v) recent antibiotic use, ***: three relevant confounders included, **: two relevant confounders included, *: ≤1 relevant confounder included.

**Abbreviations:** UTI, urinary tract infection; GV, *Gardnerella vaginalis*; HIV, human immunodeficiency virus; BV, bacterial vaginosis; RTI, reproductive tract infection; STI, sexually transmitted infections; HPV, human papillomavirus; VV, vulvovaginal; VVC, Vulvovaginal candidiasis

## Study findings

For simplicity of reporting, the following section summarizes findings from the nine highest-quality studies, selected according to the four criteria in Table 2. Studies meeting an average of three stars per criterion or a total of at least twelve cumulative stars were included. Results from all studies are available in S5 Table.

Among the nine highest-quality studies, two reported on self-reported symptoms of RTIs, STIs, or UTIs as a primary outcome, including Das 2015 who found that women using cloth had greater odds of urogenital symptoms compared to disposable menstrual pad users (aOR = 2.26; 95% CI: 1.50, 3.40) [16]. In Janoowalla 2020, researchers observed no difference in the odds of either UTI symptoms among women using single-use menstrual pads versus no menstrual pads (aOR = 1.02; 95% CI: 0.66, 1.58) or vulvovaginal symptoms comparing the same groups (aOR = 0.89; 95% CI: 0.52, 1.52) [25].

Six studies reported mixed findings on associations between menstrual products and confirmed BV, including three studies comparing various products to the use of sanitary pads. Baisley 2009 found that women using cloths, underwear, or sponges (aOR = 1.42; 95% CI: 1.02, 1.95) and women using cotton wool or toilet paper (aOR = 2.52; 95% CI: 1.21, 5.25) experienced greater odds of BV than women using sanitary pads [44]. Torondel 2018 also found that reusable cloths were weakly associated with BV infection compared to disposable sanitary pads (aPRR = 1.23; 95% CI: 1.00, 1.54) [13], and Demba 2005 found that the prevalence of BV was lower among women using sanitary pads versus traditional, cloth-based methods (43.0% vs. 50.7%; p-value = 0.32), although this difference was non-significant [34]. Two studies by Das 2015 and Das 2021 found non-significant associations between the odds of BV comparing women using silk or nylon products versus old cotton products (OR = 1.3; 95% CI: 0.9, 1.8), towels versus old cotton products (OR = 1.0; 95% CI: 0.4, 2.3), and reusable cloths versus disposable pads (aOR = 1.23; 95% CI: 0.8, 2.0) [14,16]. Finally, Mehta 2023 found that adolescent girls using menstrual cups had non-significantly lower odds of BV (aOR = 0.82; 95% CI: 0.51, 1.32) when compared to girls undergoing usual practices [19].

Among three studies assessing confirmed HIV as the primary outcome, findings were broadly inconclusive. Demba 2005 found no difference in the prevalence of HIV, comparing women using sanitary pads versus traditional, cloth-based methods (12.7% vs. 13.2%; p-value = 1.00) [34] while Zulaika 2023 found no difference in the risk of incident HIV among adolescent girls when comparing menstrual cup use (aRR = 0.88; 95% CI: 0.38, 2.05) or menstrual cup use and conditional cash transfers (aRR = 0.80; 95% CI: 0.33, 1.94) compared to usual practices alone [20]. Importantly, Phillips-Howard 2015 did observe lower prevalence of menstrual item use among women living without HIV compared to those living with HIV (60.9% vs. 73.2%; p-value < 0.01) [17].

While less frequent, studies did report on other infectious disease outcomes, including Torondel 2018 who found that reusable cloths were significantly associated with *Candida albicans* infection (aPRR = 1.54; 95% CI: 1.21, 2.00) compared to disposable sanitary pads [13]. In Mehta 2023, researchers found that menstrual cup use was associated with lower risk of composite STI outcomes (aRR = 0.77; 95% CI: 0.62, 0.95) and greater odds of vaginal microbiome compositions known to be protective against BV (aOR = 1.42; 95% CI: 1.21, 1.67) when compared to girls undergoing usual hygiene practices [19]. When assessing for other composite measures, Das 2015 found that women using reusable cloths experienced greater odds of lab confirmed BV and/or UTIs (aOR = 2.8; 95% CI: 1.7, 4.5) and lab confirmed UTIs alone (aOR = 2.0; 95% CI: 1.0, 4.0) when compared to women using disposable pads [16]. Similar, yet non-significant, findings were seen in Janoowalla 2020, when comparing odds of lab-confirmed composite UTIs (aOR = 2.09; 95% CI: 0.89, 4.91) or self-reported composite STIs (aOR = 1.86 (95% CI: 0.67, 5.20) between single-use menstrual pads versus no menstrual pads [25]. Broadly inconclusive results were reported by Das 2021, Torondel 2018, and Zulaika 2023 for vulvovaginal candidiasis, and *Trichomonas vaginalis* respectively [13,14,20].

## Discussion

There is a critical gap in the literature on menstrual product associated RUTI risks, with menstrual products and comparators seldom being fully defined. In our systematic review, only one article (3.2%) fully defined their menstrual product(s)

and just three studies (9.7%) fully defined their comparator product type across all four factors (product, comparator, out-come, and confounders) considered in our analysis [13,14,20]. Research reporting poorly defined menstrual products and comparators hinders our ability to make comparisons between studies or contexts, adds the potential for bias (e.g., length of exposure is unclear when details on frequency of product changes and/or length of menstruation are not captured), and ultimately limits our understanding of what, if any, potential risks may be associated with specific menstrual product types or characteristics. The lack of uniform product definitions and comparators does not allow for an aggregation of findings across studies which would have allowed for the conduct of a meta-analysis. Multiple studies reported "traditional" or "usual practice" as a menstrual product, which is problematic given that approaches to menstrual health vary widely across the globe [2,6,45,46]. Furthermore, our understanding of potential product-related risks is complicated by inaccu-rate labeling of study groups, such as Zulaika's cRCT where almost all girls in the "cup group" were also using pads and/or cloths, tampons, or other products [20]. Menstruators use of multiple types of products is an added challenge when trying to assess single product impact due to crossover. Lastly the labeling of menstrual products as "hygienic" or "unhy-gienic" is not especially informative, especially if broader WaSH practices are not assessed [47].

Studies evaluating tampon-associated health risks have identified various product associated risks related to specific brands and characteristics. For example, Rely tampons and toxic shock syndrome (TSS), product absorbency (e.g., 34% increased TSS risk for each additional gram of absorbency), material (e.g., vaginal mucosal drying associated with rayon polyacrylate), and length of wear (e.g., increased TSS risk with >6 hours of wear) [48,49]. These findings ultimately led to changes or removal of certain tampon products and underscore the importance of more detailed product definitions. Clear definitions of included products, comparators, and study group labeling for non-tampon menstrual products are also needed to accurately understand potential menstrual product associated risk more broadly and inform public health recommendations.

With regard to outcome definition, our review identified mixed definitions and measurement with just 54.8% of included studies defining both their RUTI outcomes and including some form of laboratory confirmation. Many RUTIs can be asymptomatic [50]. Nonetheless, nine studies relied on self-reported symptoms including non-specific or inappropriate symptom criteria increasing the risk of misclassification and negatively impacting our interpretation of potential product associated risks [15,23,24,27,30–33,42]. Examples of non-specific symptom criteria include any vaginal discharge, painful intercourse, lower abdominal pain, genital/vaginal itching, back pain, and inguinal swelling. Inappropriate RUTI symp-tom criteria included gastric ulcers, blackish discharge or passing clots during menstruation. One large cross-sectional study in India assessing menstrual product associated RTI risk, for minimally defined RTIs, also excluded 11,636 women (10.6% of their sample) who had never heard of an STI during their sampling raising concerns of sampling bias. Results from at least three studies were also at risk for recall bias, with women reporting symptoms, diagnoses, or risk factors for extended intervals ranging from three months to two years.

As for confounder consideration, less than a third (32.2%) of studies reported data on four or more confounders and/or factored them in as part of their study inclusion/exclusion criteria. Critically, nearly a fifth of studies assessing RTIs/STIs did not measure any sexual practice data despite it being an essential part of the causal pathway for these outcomes. In addition, Phillips-Howard reported that 10% of Kenyan 15-year-olds engaged in transactional sex for sanitary pads, further underscoring the importance of capturing sexual practice data for analyses assessing RUTI-menstrual product associated risk [17,18]. Without this data it is not possible to accurately assess any menstrual product associated risk for RTIs/STIs.

Among other factors, nine studies captured any data on participants' recent antibiotic use. Use of antibiotics may be higher in some settings, which may predispose some menstruators to vaginal dysbiosis [7]. A 2024 study among primarily adult Palestinian refugees in Jordan reported that almost half (43.1%) of their participants had taken prescribed or unpre-scribed prophylactic antibiotics for a RUTI in the previous three months [23]. Less than half of studies considered any personal hygiene practices such as vaginal douching despite evidence suggesting increased risks for BV, pelvic inflamma-tory disease, and possibly *Chlamydia trachomatis*, again greatly limiting the accuracy of these studies due to concerns for

unmeasured confounding [51–53]. Menstrual health and hygiene comprise more than simply the products/absorbents that are used, and relevant confounders should be measured and adjusted for as applicable.

The existing evidence evaluating the relationship between menstrual products and RUTI outcomes remains inconclusive. While several studies using self-reported symptoms of RTIs, STIs, and UTIs identified significant associations [15,16,23,26,27,30–32,39], findings were inconsistent across outcomes and product types. Null associations were frequently reported, particularly for laboratory-confirmed infections such as bacterial vaginosis, *Chlamydia trachomatis*, and *Trichomonas vaginalis* [13,14,16,18,34,35,37,38,43]. Most studies were observational and lacked the methodological rigor to support causal inference, with many relying on convenience samples from specific subpopulations, such as adolescents in school-based settings or women attending health facilities. Although two studies utilized nationally representative samples, both were restricted to women under 30 years of age, thereby excluding older menstruators and limiting broader applicability [17,32]. Notably, substantial geographic gaps remain in the literature. Of the 19 studies conducted in low- and lower-middle-income countries (LMICs), 13 were from India or Kenya, underscoring the need for expanded research across other low-income settings. Given that menstrual health practices and infection risks are shaped by diverse cultural, environmental, and epidemiological factors, findings from one setting cannot be assumed to apply across countries sharing similar World Bank income classifications.

In sum, due to widespread methodological gaps, the findings of studies that have assessed associations between non-tampon menstrual product use and RUTIs should be interpreted with caution. Furthermore, aggregating data across studies is not possible due to limitations in the outcome and comparison group definitions. Thus, it remains difficult to make generalizable statements about the risks of various RUTIs associated with different menstrual products.

There are notable strengths to our review. First, we included eligible papers published between 1984 and 2024, representing both the cumulative results of this field and the contemporary findings driving current research on menstrual products. Second, the use of two independent reviewers during the screening process minimized selection bias and improved the accuracy of study inclusion, with discrepancies resolved through discussion to ensure methodological rigor. Third, our review was not limited to English language articles, given individual co-author language skills and the use of AI to translate an article in a language not spoken by any co-authors. Fourth, we utilized a novel approach to appraise the quality of evidence for each study, accounting for both the frequency and clarity of definitions for each criterion of interest. Finally, unlike previous systematic reviews that were restricted to specific geographies [4] or focused on specific product types such as menstrual cups or reusable pads [6,54], our review encompassed studies conducted across all geographic settings and included a broader range of menstrual products.

This review also has important limitations. First, we only included publications meeting eligibility criteria from three databases (PubMed, Web of Science, and FDA MAUDE). Those indexed only elsewhere would not have been included; however, these databases likely capture most peer-reviewed literature and product safety reports relevant to our search. Second, variation in how menstrual products and infectious disease outcomes were defined across publications prevented the calculation of pooled effect size estimates. Third, our appraisal of the quality of evidence across the four criteria of evidence relied on the availability and quality of product, comparator, outcome, and confounder definitions in the main text or supplemental materials of each study. When definitions were unclear, missing, or inconsistently applied, it limited our ability to fully assess study rigor and may have introduced misclassification in our synthesis of findings. We did not contact authors for more clarification of definitions. Fourth, while AI-assisted translation was used as a pragmatic tool, we acknowledge that it may lack the nuanced accuracy of human translation. For the single article where this approach was used, the English language abstract was used to verify key details and we carefully reviewed the AI-assisted translation to confirm that the text was coherent and the scientific content was logical; however, we did not independently verify the full translation of the article with a fluent speaker. Fifth, our protocol was not registered with PROSPERO, which would have increased transparency and created an audit trail of our objectives and methodology, including search strategy and inclusion criteria.

Based on findings from this review, there are several important directions for future research clarifying the potential differential risks of RUTIs associated with various menstrual products and hygiene practices. Studies should clearly define product type (including use of ≥1 product), material, frequency of change, and washing, drying, and storage practices for reusable products. Without clear definitions, comparisons across studies will remain infeasible. Additionally, categorizing menstrual products as "hygienic" or "unhygienic" is both unnecessary and biased, particularly given the lack of consistent evidence linking specific product types to adverse health outcomes. Such labels risk reinforcing stigma rather than reflecting empirical findings and are not only biased but also significantly uninformative. Prospective observational and experimental studies are also needed to understand the causal associations between menstrual product use and infectious disease outcomes, particularly for chronic (HPV, HIV) or recurrent conditions (bacterial vaginosis, vulvovaginal candidiasis). Such studies should include robust adjustment for relevant confounders including sexual behaviors, socioeconomic status, WaSH practices, age, and antibiotic use, which have been inconsistently accounted for in prior research. While the Menstrual Practices Questionnaire (MPQ) aims to contribute to more standardized measurement, key variables that likely influence RUTI risk, such as sexual practices and menstrual cycle length, are not currently captured [55]. Next, the use of laboratory-confirmed or clinician-confirmed outcomes should be prioritized over self-reported symptoms as a proxy for RUTIs in settings where testing is appropriate and feasible. Additionally, perceptions and reporting of symptoms may vary by sex, age, cultural norms, and health literacy. Finally, if comparability across studies improves, a meta-analysis would serve as a value-add to estimate a more precise pooled effect estimate necessary to understand the broader associations between menstrual products and infectious disease outcomes.

## Conclusion

There is growing interest in understanding possible associations between menstrual products and RUTIs and safe menstrual health management. Despite the identification of methodological limitations in the field over a decade ago, contemporary studies continue to have similar issues, limiting the strength of their conclusions. We recommend that international organizations and funders focused on sexual and reproductive health develop uniform guidelines for future studies, aligning with the criteria outlined here to ensure consistency and methodological rigor. These updates could include 1) clearly defining product and comparator type, material, frequency of change, and washing, drying, and storage practices for reusable products, 2) prioritizing laboratory or clinician-confirmed outcomes over self-reported symptoms, and 3) taking relevant confounders into account including sexual behaviours, socioeconomic status, WaSH, personal hygiene practices, vaginal douching, age, and recent antibiotic use. By establishing a more rigorous evidence base the field can move towards actionable insights that advance menstrual health globally.

## Supporting information

**S1 Table. Full Literature Search Strategy for PubMed.** Adapted for Entry into Web of Science.
(DOCX)

**S2 Appendix. Data Extraction Spreadsheet.**
(XLS)

**S3 Appendix. SWiM and PRISMA Checklists.**
(DOCX)

**S4 Table. Detailed Appraisal of Quality of Evidence.**
(DOCX)

**S5 Table. Associations Between Menstrual Health Products and Infectious Disease Outcomes, Stratified by Outcome of Interest.**
(DOCX)

## Author contributions

**Conceptualization:** Ana Krause, Gregory K. Zane, Andrea C. Rivas-Nieto, Abigail Mulugeta, Jairam R. Lingappa, Stephen E. Hawes.

**Data curation:** Ana Krause, Gregory K. Zane.

**Formal analysis:** Ana Krause, Gregory K. Zane.

**Funding acquisition:** Stephen E. Hawes.

**Investigation:** Ana Krause, Gregory K. Zane, Andrea C. Rivas-Nieto, Abigail Mulugeta, Md. Rezaul Hossain, Jairam R. Lingappa, Stephen E. Hawes.

**Methodology:** Ana Krause, Gregory K. Zane, Andrea C. Rivas-Nieto, Jairam R. Lingappa, Stephen E. Hawes.

**Project administration:** Ana Krause.

**Supervision:** R. Scott McClelland, Jairam R. Lingappa, Stephen E. Hawes.

**Validation:** R. Scott McClelland, Jairam R. Lingappa, Stephen E. Hawes.

**Visualization:** Ana Krause, Gregory K. Zane.

**Writing – original draft:** Ana Krause, Gregory K. Zane, Stephen E. Hawes.

**Writing – review & editing:** Ana Krause, Gregory K. Zane, Andrea C. Rivas-Nieto, Abigail Mulugeta, Md. Rezaul Hossain, R. Scott McClelland, Jairam R. Lingappa, Stephen E. Hawes.

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
