## [Decision Letter · Decision Letter 0]

6 Oct 2025

Dear Dr. Krause,

We look forward to receiving your revised manuscript.

Kind regards,

Alison Parker

Academic Editor

PLOS ONE

Journal Requirements:

3. Please remove all personal information, ensure that the data shared are in accordance with participant consent, and re-upload a fully anonymized data set.

4. We note that there is identifying data in the Supporting Information file <Supplemental_MH_Data_Extraction_.xls, Detailed Quality Assessment.docx>. Due to the inclusion of these potentially identifying data, we have removed this file from your file inventory. Prior to sharing human research participant data, authors should consult with an ethics committee to ensure data are shared in accordance with participant consent and all applicable local laws.

-Location data

Reviewers' comments:

Reviewer's Responses to Questions

**Comments to the Author**

1. Is the manuscript technically sound, and do the data support the conclusions?

Reviewer #1: Yes

Reviewer #2: Yes

2. Has the statistical analysis been performed appropriately and rigorously?

Reviewer #1: Yes

Reviewer #2: N/A

3. Have the authors made all data underlying the findings in their manuscript fully available?

Reviewer #1: Yes

Reviewer #2: Yes

4. Is the manuscript presented in an intelligible fashion and written in standard English?

Reviewer #1: Yes

Reviewer #2: Yes

Reviewer #1: General comments

This manuscript presents a timely and important systematic review that addresses a significant gap in the literature on menstrual health. The focus on methodological quality rather than just quantitative synthesis of effects is a major strength and provides a unique contribution. The review is well-conceived, rigorously conducted, and clearly written. The findings are compelling and the recommendations are practical and actionable. The manuscript is generally of high quality and suitable for publication after minor revisions to enhance clarity and address a few methodological points.

1. Abstract: The sentence "Results were evaluated and synthesized using tabular methods according to measures of association, and across four criteria categories..." is slightly long and could be broken into two for clarity. Again, please consider stating the number of studies that fully defined products (3.2%) and comparators (9.7%) in the results to mirror the impact of the main text, rather than just the percentages for well-defined (6.5%, 9.7%).

2. Methods (i) Quality assessment framework: The framework is well-justified and operationalized. However, the "star" scoring system, while intuitive, is described in the footnote of Table 2 rather than in the main methods text. A brief description of the scoring logic (e.g., "Each criterion was scored on a scale of 1-4 stars based on the completeness of definitions...") should be included in the "Quality Assessment and Analysis" section for clarity before the reader reaches the table. (ii) Use of AI: The use of AI (ChatGPT-4) for translation is noted. This is an emerging practice. It would be prudent to add a brief sentence acknowledging this as a potential limitation (e.g., "While AI-assisted translation was used as a pragmatic tool, we acknowledge that it may not capture nuanced language as accurately as a human translator."). This could be placed in the Methods or the Limitations section. (iii) Protocol Registration: The authors state "This review was not registered." It is now considered good practice to register systematic review protocols (e.g., on PROSPERO) to minimize bias. This should be acknowledged as a limitation in the discussion.

3. Results (i) Table 2: The table is highly informative but very dense. The footnotes are essential for interpretation but are lengthy. Ensure the journal's formatting allows this critical information to be accessible to the reader. (ii) Synthesis of findings: The decision to narratively synthesize the results from the nine highest-rated studies is appropriate given the heterogeneity. The summary is balanced, highlighting both significant and null findings. The reference to S2 Table for all studies is correct. (iii) Clarity: The results for Criterion 1 (Product Definition) state "only two articles (6.5%) provided partial or full definitions across all four factors." However, in the Discussion, it's stated that "only one article (3.2%) fully defined their menstrual product(s)." This slight discrepancy (full vs. partial/full) should be clarified for consistency. It's a minor point but worth checking.

4. Discussion: As mentioned above, please add the lack of a pre-registered protocol as a minor limitation. Again, the paragraph on the "hygienic"/"unhygienic" labeling is a very important conceptual point. It could be slightly strengthened by more explicitly stating that such value-laden terms are not only biased but also scientifically uninformative.

5. Limitations: The limitations section is thorough and self-critical, covering the key points: database selection, heterogeneity preventing meta-analysis, and reliance on published data. Additionally, please incorporate the two points mentioned above: 1) the use of AI for translation, and 2) the lack of a pre-registered protocol.

6. References: Check reference #8 (OpenAI. ChatGPT-4). While citing AI models is becoming more common, ensure it aligns with the specific citation guidelines of the journal.

7. Language and clarity: Only minor proofreading is required. For example: Page 11, Line 67: "understandings" could be "understanding". Page 12, Line 92: "assistance of artificial intelligence (AI)" could be "assistance from artificial intelligence (AI)".

Reviewer #2: This is a timely and well-conducted systematic review that rigorously assesses the methodological quality of literature on a critical public health topic. The quality assessment framework and the clear, actionable recommendations for future research are significant contributions to the field.

This manuscript requires a major revision before it can be considered for publication, but all of them should be addressable.

Major Concerns

1. The authors state, "This review was not registered" (Page 6, line 134) without any justification. The pre-registration of a systematic review protocol (eg., in PROSPERO) is an important standard of practice. Registration helps prevent reporting bias by pre-specifying the review's objectives, search strategy, inclusion criteria, and analysis plan. The absence of a registered protocol must be explicitly acknowledged and discussed in the limitations section.

2. The use of AI for translating articles is an innovative approach. However, this is not yet a standard practice in systematic reviews. The manuscript lacks crucial details on the process used to ensure the accuracy of these translations. The authors should elaborate on their validation process:

a) Was the AI-translated text later reviewed by a fluent speaker of the original language? If not, can this be arranged?

b) Was a sample of articles professionally translated or reviewed to assess the reliability of the AI tool?

Without a clear validation process, the accuracy of the data extracted from these non-English articles remains questionable.

3. The four-criteria quality assessment is a key strength, but its construction requires further justification to enhance its transparency and robustness.

a) The authors should provide a rationale for weighting the four criteria (product, comparator, outcome, and confounder definition) equally. It could be argued that flaws in outcome definition (e.g., reliance on self-reported symptoms) pose a more severe threat to a study's validity than a poorly defined comparator product.

b) The thresholds for assigning stars within each criterion appear arbitrary. For example, why is the cutoff for the highest score on confounder consideration "≥4 relevant confounders"? (Table 2). A brief explanation for the choice of these cutoffs would strengthen the methodology.

4. In the "Study Findings" subsection, the authors state they are presenting "a summary of findings from the nine highest quality studies. The manuscript must clearly define how these nine studies were selected. Was a cumulative star score from Table 2 used? If so, what was the threshold for inclusion? This lack of transparency makes it difficult to evaluate the potential for selection bias.

Minor Concerns

1. The search was restricted to three databases (PubMed, Web of Science, and FDA MAUDE). The authors should provide a rationale for excluding other major health and medical databases as their omission may have resulted in missing relevant studies.

2. The star rating for confounder consideration provides a good summary but obscures important details. To provide a richer synthesis, the authors should consider a supplemental table that specifies which of the five confounder categories (sexual behavior, SES, WaSH, age, antibiotic use) were addressed in each study, rather than just presenting a summary score.

3. The authors list the inability to conduct a meta-analysis as a limitation of their review. This is more accurately a key finding of the review that powerfully demonstrates the extreme heterogeneity and poor standardization in the primary literature. It is a failure of the field, not a limitation of this review. This point should be reframed to reflect that.

4. The authors rightly criticize a study where 82-93% of participants in the "cup group" also used other products. This is an excellent point about exposure misclassification. They should consider elevating this issue in the Discussion as a critical challenge for intervention studies in this field, as concurrent product use is common and complicates the assessment of a single product's effect.

**Do you want your identity to be public for this peer review?** For information about this choice, including consent withdrawal, please see our Privacy Policy

Reviewer #1: No

Reviewer #2: No

---

## [Author Response · Author response to Decision Letter 1]

29 Oct 2025

Dear Reviewer,

Thank you so much for your careful review of our manuscript, constructive feedback and suggested edits. Please find our detailed responses to the requests for correction or information in the uploaded response to the reviewers file entitled Menstrual Health Reviewer Comments to Address and the revised manuscript file with track changes. If you require additional information or clarification please do not hesitate to let us know.

We appreciate your assistance in improving our manuscript.

Thank you kindly

---

## [Decision Letter · Decision Letter 1]

17 Nov 2025

Dear Dr. Krause,

We look forward to receiving your revised manuscript.

Kind regards,

Alison Parker

Academic Editor

PLOS ONE

Journal Requirements:

Reviewers' comments:

Reviewer's Responses to Questions

**Comments to the Author**

Reviewer #1: All comments have been addressed

2. Is the manuscript technically sound, and do the data support the conclusions?

Reviewer #1: Yes

3. Has the statistical analysis been performed appropriately and rigorously?

Reviewer #1: Yes

4. Have the authors made all data underlying the findings in their manuscript fully available?

Reviewer #1: Yes

5. Is the manuscript presented in an intelligible fashion and written in standard English?

Reviewer #1: Yes

Reviewer #1: General comments

This is a highly relevant, well-conducted, and timely systematic review. It addresses a critical gap in women's health by rigorously assessing the methodological quality of the existing literature on a topic of significant public health interest. The manuscript is generally well-written, the methods are sound and clearly described, and the conclusions are justified by the results. The primary strength lies in its novel quality assessment framework, which effectively diagnoses the key weaknesses preventing definitive conclusions in this field.

Specific Comments

1. Abstract: The sentence in the Methods, "Results were evaluated and synthesized using tabular methods according to measures of association. This was done across four criteria categories..." is slightly awkward. Consider combining for flow: "Results were evaluated and synthesized using tabular methods according to measures of association, assessed across four criteria categories:..."

2. Language and Grammar: The manuscript is well-written and clear. The language is formal and appropriate for a scientific journal. Only very minor proofreading issues were noted, which are likely to be caught in the final copyediting process. (i) Page 11, Line 67: "rigorous understandings" could be changed to "a rigorous understanding" for better flow. (ii) Page 42, Line 94: "assistance fromof artificial intelligence" – the tracked change has left an artifact. This should be "assistance from artificial intelligence".

**Do you want your identity to be public for this peer review?** For information about this choice, including consent withdrawal, please see our Privacy Policy

Reviewer #1: No

---

## [Author Response · Author response to Decision Letter 2]

18 Nov 2025

Dear Reviewers,

Thank you very much for your detailed review of our manuscript and the suggested edits. Please find our detailed responses to the requests for correction in the uploaded response to the reviewers file entitled Menstrual Health Reviewer Comments to Address_2 with changes made in the updated version of the manuscript.

Thank you kindly,

Ana

---

## [Editor Report · Decision Letter 2]

26 Nov 2025

Assessing the relationship between menstrual products and reproductive and urogenital tract infections (RUTIs): a systematic review evaluating the evidence and recommendations for future research

PONE-D-25-45914R2

Dear Dr. Krause,

We’re pleased to inform you that your manuscript has been judged scientifically suitable for publication and will be formally accepted for publication once it meets all outstanding technical requirements.

Kind regards,

Alison Parker

Academic Editor

PLOS ONE
---

## [Editor Report · Acceptance letter]

PONE-D-25-45914R2

PLOS One

Dear Dr. Krause,

I'm pleased to inform you that your manuscript has been deemed suitable for publication in PLOS One. Congratulations! Your manuscript is now being handed over to our production team.

Kind regards,

on behalf of

Dr. Alison Parker

Academic Editor

PLOS One